# Model simulations of cooking organic aerosol (COA) over the UK using estimates of emissions based on measurements at two sites in London

Riinu Ots[1,2], Massimo Vieno[2], James D. Allan[3,4], Stefan Reis[2,5], Eiko Nemitz[2], Dominique E. Young[3,*], Hugh Coe[3], Chiara Di Marco[2], Anais Detournay[2], Ian A. Mackenzie[6], David C. Green[7], and Mathew R. Heal[1]

[1]School of Chemistry, University of Edinburgh, Edinburgh, UK
[2]Natural Environment Research Council, Centre for Ecology & Hydrology, Penicuik, UK
[3]School of Earth, Atmospheric and Environmental Sciences, University of Manchester, Manchester, UK
[4]National Centre for Atmospheric Science, University of Manchester, Manchester, UK
[5]University of Exeter Medical School, European Centre for Environment and Health, Knowledge Spa, Truro, UK
[6]School of GeoSciences, University of Edinburgh, Edinburgh, UK
[7]MRC PHE Centre for Environment and Health, King's College London, London, UK
[*]now at: Department of Environmental Toxicology, University of California, Davis, CA, USA

*Correspondence to:* M. Heal (M.Heal@ed.ac.uk) and R. Ots (R.Ots@ed.ac.uk)

**Abstract.**

Cooking organic aerosol (COA) is currently not included in European emission inventories. However, recent positive matrix factorization (PMF) analyses of aerosol mass spectrometer (AMS) measurements have suggested important contributions of COA in several European cities. In this study, emissions of COA were estimated for the UK, based on hourly AMS measure-

5  ments of COA made at two sites in London (a kerbside site in central London and an urban background site in a residential area close to central London) for the full calendar year of 2012 during the Clean Air for London (ClearfLo) campaign. Iteration of COA emissions estimates and subsequent evaluation and sensitivity experiments were conducted with the EMEP4UK atmospheric chemistry transport modelling system with a horizontal resolution of 5 km × 5 km.

The spatial distribution of these emissions was based on workday population density derived from the 2011 census data. The

10 estimated UK annual COA emission was 7.4 Gg per year, which is an almost 10% addition to the officially reported UK national total anthropogenic emissions of $PM_{2.5}$ (82 Gg in 2012), corresponding to $320 \, \mathrm{mg \, person^{-1} \, day^{-1}}$ on average. Weekday and weekend diurnal variation in COA emissions were also based on the AMS measurements. Modelled concentrations of COA were then independently evaluated against AMS-derived COA measurements from another city and time period (Manchester, Jan–Feb 2007), as well as with COA estimated by a chemical mass balance model of measurements for a two-week period at

15 the Harwell rural site (~80 km west of central London).

The modelled annual average contribution of COA to ambient particulate matter (PM) in central London was between 1–2 $\mathrm{\mu g \, m^{-3}}$, and between 0.5–0.7 $\mathrm{\mu g \, m^{-3}}$ in other major cities in England (Manchester, Birmingham, Leeds). It was also shown that cities smaller than London can have a central hot-spot of population density of smaller area than the computational grid cell, in which case higher localised COA concentrations than modelled here may be expected.



Modelled COA concentrations dropped rapidly outside of major urban areas (annual average of $0.12\,\mu g\,m^{-3}$ for the Harwell location), indicating that although COA can be a notable component in urban air, it does not have a significant effect on PM concentrations on rural areas.

The possibility that the AMS PMF-apportionment measurements overestimate COA concentrations by up to a factor of 2 is
discussed. Since COA is a primary emission, any downward adjustments in COA emissions would lead to a proportional linear downward scaling in the absolute magnitudes of COA concentrations simulated in the model.

## 1   Introduction

Airborne particulate matter (PM) has multiple impacts on atmospheric processes. It affects the transport, transformation and deposition of chemical species and influences radiative forcing (Pöschl, 2005; USEPA, 2009). Ambient surface concentrations
of PM in particular contribute to substantial adverse human health effects (Heal et al., 2012; Lim et al., 2012; WHO, 2013; Brauer et al., 2016). The carbonaceous component constitutes a substantial fraction of total particle mass (USEPA, 2009; Putaud et al., 2010; AQEG, 2012), and arises through many diverse primary emission sources and in situ atmospheric processes (Fuzzi et al., 2006; Hallquist et al., 2009; Jimenez et al., 2009). It is necessary to accurately apportion the origin of organic aerosol (OA) in order to devise effective mitigation of ambient PM. This can be facilitated through the integration of measurements
and modelling.

Even allowing for the uncertainties in defining and measuring OA components, current atmospheric chemistry transport model (ACTM) simulations tend to underestimate observed amounts of OA (Simpson et al., 2007; Murphy and Pandis, 2009; Hodzic et al., 2010; Aksoyoglu et al., 2011; Jathar et al., 2011; Bergström et al., 2012; Koo et al., 2014; Prank et al., 2016). In some cases, this underestimation has been shown to be due to missing or under-represented emission sources in the underlying
emission inventories (Simpson et al., 2007; Denier van der Gon et al., 2015). One such primary source of OA is cooking organic aerosol (COA).

In the US, emissions of OA from meat charbroiling (grilling) or frying have been known for decades to be a significant contributor to ambient air quality (Rogge et al., 1991; Hildemann et al., 1991). Consequently, cooking aerosol is included as a component of particulate matter in the US national emission inventory (USEPA, 2004). In Europe, the impact of cooking
emissions on ambient air quality via national emissions has so far been neglected. This might be because of an assumption that there is less meat charbroiling in Europe than in the US. However, using positive matrix factorization (PMF) analyses of aerosol mass spectrometer (AMS) measurements, several recent European studies have apportioned a substantial part of submicron OA to cooking. Allan et al. (2010) estimated that the average contribution of COA to OA in Manchester, UK, was 19% whilst in London, UK, it was 22–30%. For Barcelona, Spain, Mohr et al. (2012) reported a 17% contribution to OA from
COA, and measurements at different sites in Paris, France, were interpreted as indicating a 15–20% average contribution from COA (Crippa et al., 2013a, b). Allan et al. (2010) also reported that the COA in London is more likely to be produced from vegetable seed oils used during frying, rather than solely from meat cooking.





Based on the aforementioned PMF apportionment measurements of OA components in Paris, Fountoukis et al. (2016) estimated the emissions of COA to be ~80 $\mathrm{mg\,person^{-1}\,day^{-1}}$ on average. Adding these emissions to their model based on population density enabled their simulations to reproduce measured COA concentrations at two sites during the MEGAPOLI campaign. Fountoukis et al. (2016) then added the same 80 $\mathrm{mg\,person^{-1}\,day^{-1}}$ emission of COA to their model for a European domain, and concluded that, based on this estimate, the contribution of COA emissions from other countries to COA concentrations in Paris was between 0.1–0.2 $\mathrm{\mu g\,m^{-3}}$ of $\mathrm{PM_1}$. Discussion of potential uncertainties in the quantification of COA by PMF of AMS measurements is presented later in this paper.

In this work, AMS-derived measurements of COA for a full calendar year at two sites in London during the 2012 Clean Air for London campaign (ClearfLo; Bohnenstengel et al. (2014); Young et al. (2015)) were combined with gridded UK population density data (Reis et al., 2016) to construct estimates of COA emissions across the UK. The EMEP4UK ACTM (Vieno et al., 2010, 2014, 2016; Ots et al., 2016) was then applied to conduct calibration tests of these novel gridded and temporally-variable emissions of COA, and predictions were compared with a third, independent, dataset of measurements of COA made by AMS in Manchester in Jan–Feb 2007 (Allan et al., 2010).

## 2   Methods

### 2.1   Model description

The EMEP4UK model is a regional application of the EMEP MSC-W (European Monitoring and Evaluation Programme Meteorological Sythesizing Centre-West) model. The EMEP MSC-W model is a 3-D Eulerian model that has been used for both scientific studies and to support policy making in Europe. A detailed description of the EMEP MSC-W model, including references to evaluation and application studies is available in Simpson et al. (2012), Schulz et al. (2013), and at www.emep.int. The model used here was based on version v4.5.

The model has 21 vertical levels, extending from the surface to 100 hPa. The lowest vertical layer is ~40 m thick, and the horizontal resolution used in this study is 5 km × 5 km over a British Isles domain. The model uses one-way nesting from an extended European domain (simulated with 50 km × 50 km horizontal resolution), but this has no bearing on the COA concentrations presented in this study as COA emissions are not compiled for European countries and in this work were only implemented for the UK. The model was driven by output from the Weather Research and Forecasting (WRF) model (www.wrf-model.org, version 3.1.1) including data assimilation of 6-hourly model meteorological reanalysis from the US National Center for Environmental Prediction (NCEP)/National Center for Atmospheric Research (NCAR) Global Forecast System (GFS) at 1° resolution (NCEP, 2000).

The performance of this version of the EMEP4UK model simulating a standard suite of gas-phase components and secondary inorganic aerosol PM components is reported in Ots et al. (2016) comparing with a full year of measurements in London in 2012.

For the present study, a COA tracer was added into the model with dry and wet deposition properties similar to other fine (i.e. $\mathrm{PM_{2.5}}$) primary OA components (see Simpson et al. (2012) for aerosol specifications in the EMEP MSC-W model). This



COA tracer is non-volatile and does not undergo chemical ageing, but it is included in the total OA budget for the absorptive partitioning of secondary organic aerosol species.

## 2.2  AMS measurements used in this study

The construction of COA emissions estimates were based on measurements made during the ClearfLo project (Bohnenstengel et al., 2014) at two sites in London, shown in Fig. 1. Marylebone Road is a 'kerbside' site on the edge of a heavily-trafficked urban through-road, whilst the North Kensington site is classified as urban background and is situated in the carpark of a school. The measurements at Marylebone Road were taken with a Q-AMS (Quadrupole AMS; Jayne et al. (2000)) between 11-Jan-2012 and 1-Feb-2013 and were averaged to hourly values, yielding 5996 data points (Detournay et al. (2015); several gaps in the measured data were caused by problems with the instrument computer). The measurements at North Kensington were taken with a cToF-AMS (compact Time of Flight AMS; DeCarlo et al. (2006)) between 11-Jan-2012 and 23-Jan-2013, and with a HR-ToF-AMS between 21-Jul-2012 and 19-Aug-2012 (High-Resolution ToF-AMS), hourly averaging yielded 8035 data points (Young et al., 2015). The annual average (for 2012) concentrations of COA derived from the AMS measurements were 2.2 $\mu g\,m^{-3}$ at Marylebone Road, and 0.8 $\mu g\,m^{-3}$ at North Kensington. Figure S1 shows a satellite image of the Marylebone Road measurement site with food-related commercial establishments (cafes, restaurants, etc.), as known to Google, marked. (The accuracy or comprehensiveness of these establishments marked on Google Maps has not been verified, but are presented to illustrate the number of food outlets in the area.) There is no direct source of cooking emissions close to the Marylebone Road measurement site, so the measured concentrations, although high, are likely to represent an average of the many COA emissions sources in the vicinity.

Positive matrix factorisation (PMF) seeks to reproduce the measured time series of the organic mass spectrum through a linear composition of a (user-selectable) number of factor spectra (representing different OA types or sources) and their mass contribution, taking into account the precision associated with each measurement. Subjectivity is minimized by comparison of concentration time-series with independent measurements and assessment of the robustness of the solution, e.g. through boot-strapping. COA has been identified as a contributor to urban OA measurements because it exhibits a distinct diurnal cycle and the associated factor spectrum is very similar to that of lab-generated cooking oil aerosol (Allan et al., 2010). Nevertheless, there are some inherit uncertainties involved in deriving COA concentrations with AMS measurements. For example, AMS measurements need to be corrected for the fraction of aerosol that is not effectively vaporised due to bounce from the hot surface involved in the AMS's detection mechanism. Whilst this is well characterised for typical, internally-mixed ambient aerosols (e.g. Middlebrook et al. (2012)), it is possible that the COA measured by the AMS is not well mixed with other aerosol components and could therefore be detected at a higher efficiency. If this were the case, AMS measurements may overestimate COA concentrations by up to a factor of 2.

Indeed, a study comparing AMS-PMF derived concentrations of PM components with those estimated based on measurements and a chemical mass balance (CMB) model at the North Kensington site during a 2-week period in the same campaign used in this study concluded that AMS derived COA was on average 1.6 times higher than the CMB derived values, but good correlation was seen (a linear fit of $AMS_{COA} = 2.24 \times CMB_{COA} - 0.33$, with $r = 0.89$), Yin et al. (2015)), which is consis-





tent with the AMS collection efficiency (CE) being higher than the usual 0.5. There are also additional sources of uncertainty with PMF, in particular rotational ambiguity, which can cause both over- and underestimates (Allan et al., 2010; Paatero et al., 2002). However, the CMB approach is also not without its uncertainties, in particular that the COA marker(s) used in the CMB may not be fully representative, and because of the need to scale marker concentration to COA concentration.

In summary, the full quantification of COA by AMS (and any other approach) requires further research but it is currently more likely that the AMS overestimates the COA than underestimates it.

### 2.3 Spatial distribution of COA emissions

Figure 1a shows the residential population density data in the central London area at 1 km × 1 km resolution, overlaid by the EMEP4UK grid cells (5 km × 5 km), and Fig. 1b the equivalent workday population[1] density. These datasets were compiled

by Reis et al. (2016) based on the 2011 UK Census, with population data provided on output area level, spatially distributed on a 1 km × 1 km grid for England, Wales and Northern Ireland using the Land Cover Map 2007 land use classes 'urban', 'suburban' and 'urban industrial'.

The North Kensington and Marylebone Road measurement sites are situated in different model grid cells. The Marylebone Road grid cell includes most of the very central part of London, with many popular tourist attractions such as Madam Tussauds,

Buckingham Palace, Big Ben and the Houses of Parliament, and the London Eye. Even though there are no gridded data of 'tourist population density', the workday population density data include indication for tourist numbers as many of the jobs (and therefore the workday population density) in this area will be directly related to, or indirectly dependent on, the tourism sector. The total workday population for the grid cell of the Marylebone Road site is more than 3 times higher than for the grid cell for the North Kensington site. The residential population density in the North Kensington grid cell, however, is higher than

in the Marylebone Road grid cell. The annual-average measured COA concentration at the Marylebone Road site was 2.8 times higher than at the North Kensington site, very similar to the ratio in gridded workday population density. Therefore, workday population density was chosen as the spatial distribution weighting to apply to COA emissions in the model input.

At present, gridded workday census data are only available for England, Wales, and Northern Ireland, so for Scotland the residential population data had to be used instead. The finer resolution (1 km) information of the COA emissions gridded to

population density data was aggregated appropriately to the coarser model resolution during input data preparation.

### 2.4 Annual total emitted COA

Based on sensitivity tests (Table 1), the annual total COA emissions for the UK applied to the model was set to 7.4 Gg. (The spatial distribution applied to these emissions is explained in the previous section, the temporal variation is explained in the following section.) This is a 9% addition to the UK national total $PM_{2.5}$ emissions for the year 2012 (82 Gg, NAEI (2013)).

---

[1]The *workday population* is a redistribution of the usually resident population to their place of work, while residents who are not in work remain at their area of residence. The workday population of an area is defined as *"all usual residents aged 16 and above who are in employment and whose workplace is in the area, and all other usual residents of any age who are not in employment but are resident in the area"*; Source: Office for National Statistics, http://www.ons.gov.uk/ons/index.html




This emission corresponds to about $320 \, \mathrm{mg \, person^{-1} \, day^{-1}}$ (for a population of 63 million), which is 4 times higher than estimated by Fountoukis et al. (2016) for France. This difference might be explained by differences in cuisines - it is possible that relatively more grilled, fried and, in particular, deep-fried food is consumed in the UK than in France. Furthermore, it is also possible that the difference in the measurement site locations relative to the very centre of either megacity, and the

representativeness of the measurement location to model grid average, could increase or decrease the estimate made for the whole country.

## 2.5 Temporal variation of COA emissions

The average diurnal profiles of measured COA concentrations the Marylebone Road and North Kensington sites are shown in Figs. 2a and 2b. The measured diurnal cycle of COA concentrations at Marylebone Road was taken as a basis for a temporal

emission profile. Marylebone Road was chosen because the concentrations are substantially higher than at North Kensington and show a stronger diurnal variation with more pronounced peaks around both the lunchtime (12:00–14:00) and evening (dinnertime, 18:00–21:00) periods. Even though the diurnal COA concentration variation at Marylebone Road is clearly driven by these meal times, it is further influenced by atmospheric processes such as changing boundary layer height and dispersion, potentially introducing a non-linearity between emissions and concentrations. Therefore, the ACTM was used to assess these

processes using sensitivity runs with different diurnal emission profiles. As a first test, the diurnal profile of COA emissions was set exactly to the measured profile at Marylebone Road, with separate profiles for weekdays and weekend days (the lunchtime peak is more pronounced on weekdays than on weekends). Further sensitivity runs with modified diurnal emission profiles were conducted with the goal of optimising modelled-measured agreement simultaneously at both the Marylebone Road and North Kensington measurement sites. These sensitivity runs and the final diurnal weekday and weekend diurnal emission

profiles selected are explained in detail in the Supplementary Information. The emissions total was applied to all seven days of the week because the measurements showed only very small day-of-the-week trends (Fig. 2c and d) and differed between the two measurement sites. No seasonality (or monthly) variation was assigned to the emission profile under the assumption that cooking is a consistent year-round activity. It is, however, recognised that cooking emissions may also be strongly affected by tourist population density and may thus have some degree of seasonality. For example, the 2012 Summer Olympics took place

in London from 25-July to 12-August attracting 680,000 overseas tourists alone (UK Office for National Statistics, 2012).

## 2.6 Summary of the newly composed COA emissions

– The emissions were spatially gridded to workday population density, not residential population density, as this captured the relative difference between observed annual average COA concentrations between the central, commercially-based (Marylebone Road site) and the residential (North Kensington site) areas.

– The annual total COA emission for the UK was based on a series of sensitivity runs to minimise total bias for both sites. The final amount was 7.4 Gg per year, which is an almost 10% addition to the officially reported total $PM_{2.5}$ emissions (82 Gg in 2012). This corresponds to about $320 \, \mathrm{mg \, person^{-1} \, day^{-1}}$ on average.



- The diurnal profile of COA emissions (i.e. the relative increase of emissions during lunch or dinner) was mainly based on the observations at Marylebone Road (as the concentrations were higher and the emission profile was therefore more pronounced at the very central location). Slightly different diurnal cycles were assigned to weekday and weekend COA emissions, but no day-of-the-week or monthly variations were applied to the emissions.

The annual gridded UK COA emissions used in the model simulations are shown in Fig. 3a, and the resulting annual-average modelled COA surface concentrations (for 2012) are shown in Fig. 3b.

### 2.7    Model evaluation statistics used in this study

The following numerical metrics were used for model evaluation: FAC2 (Factor of 2) - the proportion of modelled concentrations that are within a factor of 2 of the measured concentrations; NMB - normalised mean bias; NMGE - normalised mean
gross error, which is defined as:

$$NMGE = \frac{\frac{1}{n}\sum_{i=1}^{n}|M_i - O_i|}{\overline{O}}, \tag{1}$$

where $M_i$ is the $i$th modelled value, $O_i$ is the corresponding measured value, $\overline{O}$ is the mean measured value, and $n$ in the total number of observations; $r$ - correlation coefficient; and COE - coefficient of determination, which is defined as:

$$COE = 1.0 - \frac{\sum_{i=1}^{n}|M_i - O_i|}{\sum_{i=1}^{n}|O_i - \overline{O}|}. \tag{2}$$

A COE of 1 indicates perfect agreement between model and measurements. Although the COE does not have a lower bound, a zero or negative COE implies that the model cannot explain any of the variation in the observations (Legates and McCabe, 2013).

### 3    Results and Discussion

The results section is organised as follows. First, hourly concentrations and average diurnal profiles of measured and modelled
COA at the two sites in London are evaluated. Second, an evaluation of daily-averaged measured and modelled COA is presented. These analyses are undertaken for the same sites that were used to estimate the COA emissions. In the third part of the results section, the modelled concentrations are evaluated against a separate, short (two-week) period of measurements from a different location, the centre of the city of Manchester. Finally, modelled concentrations of COA in other major UK cities, as well as in the vicinity of London are discussed.

### 3.1    Hourly comparison of measured and modelled COA concentrations in London

The average hourly profiles (diurnal cycles) of measured and modelled COA concentrations at the Marylebone Road and North Kensington sites are shown in Fig. 2a and b, respectively. As explained above, the diurnal COA emission profile applied to the model was mainly based on measurements at the Marylebone Road site. Since COA measurements at this site had a notable



lunchtime peak, the modelled lunchtime peak at North Kensington (12:00–14:00, Fig. 2b) is slightly elevated compared with the measurements, but, overall, measured and modelled diurnal cycles are in very good agreement ($r = 0.99$ for Marylebone Road; $r = 0.93$ for North Kensington).

Scatterplots of modelled and measured hourly COA concentrations at the Marylebone Road and North Kensington sites, with weekdays and weekends separated, are shown in Fig. 4 (the time series of these hourly data are shown in Figs. S6–S9). The average concentrations for each panel of Fig. 4 are given in Table 2. At the Marylebone Road site, neither the hourly evaluation statistics, nor the mean COA concentrations, show a difference between weekdays and weekends. However, differences in the statistics are observed between weekdays and weekends at the North Kensington site: mean COA concentration for weekdays is $0.7 \, \mu g \, m^{-3}$, whereas for weekends it is $1.1 \, \mu g \, m^{-3}$. As no day-of-the-week variation was applied to total daily emissions (only to the weekday/weekend diurnal emission profiles), the model can not reproduce this difference (both weekday and weekend mean simulated COA concentrations are $0.9 \, \mu g \, m^{-3}$). It is possible in the model to give emissions from each source sector a weekly cycle. This is done for several sectors already. For example, road transport emissions are higher during weekdays, whereas residential heating emissions are higher during the weekends. For cooking emissions, a weekly cycle might be justified for more office dominated areas (like the North Kensington area), but not for the very central commercial and recreational area where the Marylebone Road site is located. It is possible that central London is an exception and that overall, it would be better to assign a weekly cycle to emissions from cooking activities (as it is possible that in every other city than the capital, weekends are busier than weekdays in terms of eating out and therefore a day-of-week factor would be justified). Therefore, more measurements (or alternatively, statistics about the spatial and temporal variability of restaurant customer numbers during different days of the week) should be collected.

Overall, the hourly evaluation statistics are similar for both sites (Fig. 4): FAC2 is 62% (weekdays) and 55% (weekends) for Marylebone Road, and 62% (weekdays) and 65% (weekend) for North Kensington; NMGE is 69% and 60% for Marylebone Road and 64% and 52% for North Kensington; $r$-values are 0.46 and 0.56 for Marylebone Road and 0.53 and 0.63 for North Kensington. The conclusion is that the diurnal emission profiles derived as model input for COA emissions result in similar model performance for both types of area.

## 3.2 Evaluation of daily-averaged COA concentrations in London

Time series of daily averaged modelled and measured COA concentrations along with daily evaluation statistics for the two sites in London are shown in Fig. 5. Based on the hourly evaluation in the previous section, some disagreement can be expected at the North Kensington site by not including in the model any difference between weekday and weekend emissions. Despite this, it was shown that the hourly evaluation statistics were similar for both sites. However, North Kensington and Marylebone Road show very different results for the daily evaluation. For the North Kensington site, daily performance is satisfactory (Fig. 5a), with an $r$-value of 0.56 and a COE of 0.19. The NMGE of 43% could be attributed to the uncertainties in the COA emissions (including the weekdays vs weekends difference), as well as uncertainties in the meteorological driver. For Marylebone Road on the other hand (Fig. 5b), the model does not satisfactorily simulate the measured daily variation of COA concentrations ($r = 0.11$, COE = -0.22).



Figures 6a–d show polar plots of measured and modelled COA concentrations for the North Kensington and Marylebone Road sites. Wind data are from the the Heathrow Airport meteorological station (Met Office, 2012), about 20 km to the west of central London. Meteorological observations from the airport, rather than more local measurements, are used as the airport measurements are unaffected by large buildings and are likely to be more representative of larger scale wind over Greater London. For comparability, the same wind data are used for both measured and modelled concentrations. Furthermore, the days with missing measurements (Fig. 5, especially important for the Marylebone Road site) are also removed from the modelled concentrations polar plots. However, it should be noted that the datasets used in these plots still differ in size between the two sites ($n$ days = 191 at Marylebone Road and $n$ days = 340 at North Kensington).

It can be seen from Figs. 6a and 6b that at the North Kensington site both measurements and model show higher concentrations when the wind is from the east. This is expected as North Kensington is slightly to the west of central London (Fig. 1) and therefore wind from the east has passed over more local emission sources. However, the polar plots for Marylebone Road show substantial differences between measured and modelled concentrations. The model simulates higher daily COA concentrations at lower wind speeds from all directions (Fig. 6d, see Fig. S10 for scatterplots of these values conditioned by four divisions of wind directions). In contrast, the measurements show a gradient of higher concentrations when winds are southerly and lower concentrations for northerly winds (Fig. 6c, see Fig. S11 for scatterplots of these values conditioned by wind speed quantiles). A detailed map of the Marylebone Road location is shown in Fig. 7. There is a large park (Regent's Park) just to the north of the Marylebone Road measurement site, explaining why lower concentrations are measured from that direction. The model can not of course resolve this 'sub-grid' variation (the model's horizontal resolution is 5 km×5 km, as shown in Fig. 1) and thus misses the effects of the park. Whilst the use of the synoptic wind from Heathrow Airport will represent medium to far-field influences more accurately, the funnelling of the air flow by the street canyon will affect the contribution from very local sources and the degree of ventilation vs. build-up of material emitted from within the canyon. These effects are likely to lead to a more variable concentration at the Marylebone Road roadside site than at the North Kensington background site. Measurements at different locations and more modelling studies (including different models, for example an urban dispersion model) of COA concentrations in London, as well as in other cities would be necessary to draw further conclusions about the variability of COA concentrations in a street canyon situation.

### 3.3 Comparison with COA measurements in Manchester in 2007

In this section, modelled concentrations (using the emissions based on measurements in London, 2012) are compared with a two-week period of AMS and PMF apportionment measurements in Manchester, Jan–Feb 2007 (taken with a cToF-AMS; Allan et al. (2010)). The Manchester measurement site location, as well as gridded workday population density (1 km × 1 km resolution) overlaid with the modelling grid (5 km × 5 km) is shown in Fig. 8. The model grid cell in which the measurement site is situated includes an area of a few km in width where the workday population density is several times higher than in the rest of the 5 km × 5 km cell (this is very central Manchester around the main train station). Since the measurement site was also located in this high workday population density area it is likely that the measured concentrations represent the highest COA concentrations in Manchester, in contrast, the model simulates an average concentration for the whole grid cell which will





be lower than at the sub-grid measurement hot-spot. It should also be noted that the Manchester measurement site is located 0.5 km from a 'Chinatown', which could have a direct influence on the measured COA concentrations due to its high number of restaurants and deep-drying.

The time series of hourly-averaged measured and modelled concentrations during the 2-week period of measurements in
Manchester are shown in Fig. 9a. Average diurnal cycles are shown in Fig. 9b, and a scatterplot of daily averaged concentrations in Fig. 9c. Modelled concentrations are a factor of 2 lower than measurements (NMB = -50%), likely due to the sub-grid modelling issue discussed above. Nevertheless, there is very good measurement-model correlation ($r = 0.80$ for diurnal profiles, $r = 0.63$ for hourly-averaged concentrations, $r = 0.86$ for daily-averaged concentrations). This indicates that the diurnal profile for COA emissions derived based on measurements in London is also suitable for use in other areas. However, the results
suggest that because London is a megacity, the high concentrations in the central area can on average be captured by simulations with the 5 km × 5 km horizontal resolution, whereas for Manchester, a finer set-up (~1-2 km for example) would be needed. Nevertheless, the modelled concentrations are still useful in representing the spatially-averaged concentrations within the whole grid cell. Even allowing for the model resolution, the negative bias between model and measurement suggests that the per capita emissions estimate for COA derived from the London measurements is not an overestimate for COA emissions in Manchester
(setting aside the discussion that both London and Manchester AMS measurements maybe be overestimates of COA).

### 3.4 Maximum modelled COA concentrations in London, Manchester, Leeds, and Birmingham

Some statistics for the range of daily-average COA concentrations at the two London sites are given in Table 3. The modelled and measured mean values match closely, with a bias of -0.1 $\mu g\,m^{-3}$ for the Marylebone Road site, and +0.1 $\mu g\,m^{-3}$ for the North Kensington site. For the Marylebone grid cell, two sets of statistics of modelled concentrations are given: one matched
for data availability with measurements (i.e. missing January, most of March, June and July, other odd days), and one for the full calendar year. The influence of the missing periods is small in this case (full year mean is 2.0 $\mu g\,m^{-3}$, measurements-matched mean is 2.1 $\mu g\,m^{-3}$).

The model grid cell encompassing the Marylebone Road site has the highest annual average modelled COA concentration in London, and indeed across the whole of the UK. Therefore, these statistics (both measured and modelled) likely represent
the maximum contribution cooking emissions might have on a 5 km × 5 km area. The annual average COA concentration of 2 $\mu g\,m^{-3}$ in central London is relevant as that constitutes 20% of the WHO $PM_{2.5}$ air quality guideline of 10 $\mu g\,m^{-3}$ for example.

Figure 10 shows the time series of daily-averaged modelled concentrations for 2012 for the other most populous cities in the the UK - Birmingham, Manchester, and Leeds (Glasgow is omitted as the workday population data were not yet available
for Scotland). The data shown are for the grid cell over these cities with the largest annual-average COA concentrations. The higher COA concentrations in these cities are also visible in the annual average map of modelled COA surface concentrations in Fig. 11b. Based on the gridded workday population density in Manchester and the results shown in the previous section, it is likely that these simulated 5 km × 5 km concentrations do not capture the central hot-spots of cities smaller than London, but capture the average of an area wider than the centre itself.





As an annual average in 2012, modelled COA contributed 0.5–0.7 µg m$^{-3}$ in these cities (data given in Fig. 10). On 36 days of 2012 (90th percentile, denoted Up10 in Fig. 10), modelled COA concentrations are over 0.9 µg m$^{-3}$ in Leeds and Birmingham, and over 1.3 µg m$^{-3}$ in Manchester. As a 95th percentile of daily averages for 2012, modelled COA contributed 1.3, 2,2 and 2.9 µg m$^{-3}$ in Leeds, Birmingham and Manchester, respectively.

## 3.5 COA concentrations in the vicinity of London

The map of UK modelled surface concentrations of COA presented in Fig. 3 shows that the impact of cooking emissions on an annual average basis is spatially very limited, as COA concentrations drop markedly outside the highly populated urban areas. There are no PMF apportionment measurements of COA concentrations reported outside UK urban areas, but daily-averaged modelled concentrations (for 2012) at Harwell are shown in Fig. 12a for an illustration of anticipated non-urban

COA concentrations (Harwell is an EMEP supersite ~80 km west of central London, its exact location is marked on maps in Fig. S8). Harwell was also a measurement site during the ClearfLo project. The modelled time series indicate that the COA concentrations at Harwell are relatively small and episodic. In fact, their characteristic diurnal signature is entirely lost (Fig. 12b) and their time-series becomes very similar to that of other emissions dominated by population density. This is the reason why PMF commonly fails to resolve COA and hydrocarbon-like organic aerosol (HOA, dominated by vehicular

emissions) at rural sites.

The modelled COA concentrations for Harwell are similar to the COA concentration derived by Yin et al. (2015) with the chemical mass balance (CMB) method for the same site. For the period 12-Jan-2012 to 8-Feb-2012 Yin et al. (2015) estimate COA of 0.13 µg m$^{-3}$ (note text in this paper also refers to a COA average value of 0.12 µg m$^{-3}$); the model here yields a concentration of 0.17 µg m$^{-3}$ for the same period, 0.12 µg m$^{-3}$ for the full year average.

Modelled surface concentrations near the Greater London area for the 18 highest days (95th percentile: 0.43 µg m$^{-3}$ for Harwell) are shown in Fig. S8. Most of the higher concentrations at these location come from London, with the exception of 11-Feb and 12-Feb, when some traces of COA concentrations arrive from northern England. Furthermore, as even the 95th percentile of daily averaged COA concentrations in the vicinity of London sites is rather low, compared with the COA concentrations experienced within the large urban areas, this demonstrates that the impact of cooking emissions is also spatially

very limited on a daily basis.

# 4 Conclusions

In this study, spatially resolved estimates of emissions of cooking organic aerosol (COA) which are currently not included in European emissions inventories were generated for the UK. The magnitude and spatial and diurnal distributions of COA emissions have been derived from determinations of COA concentrations by positive-matrix factorisation (PMF) of aerosol

mass spectrometer (AMS) measurements at two sites in London for the full calendar year 2012 (Marylebone Road, a kerbside site in central London; and North Kensington, an urban background site in a residential area close to central London).




An evaluation of daily concentrations in London revealed different results for the two sites. For the North Kensington site, the model captured day-to-day variability throughout the year ($r = 0.56$, COE = 0.19), whereas for the Marylebone Road site, the model could not simulate observed inter-day variability ($r = 0.11$, COE = -0.22). Based on polar plots of measured wind directions, the likely source of this disagreement is a sub-(model)-grid effect at the Marylebone Road site and local air flows.

Comparing model results with measurements for another time period and location (Manchester, Jan–Feb 2007) suggests that the diurnal profile of COA emissions derived from 2012 measurements at Marylebone Road is suitable for simulating COA concentrations at other central urban areas.

It is shown that in London, annual average COA concentrations are between 1–2 $\mu g\,m^{-3}$ (urban background site to urban central site). Both the measurements and modelled concentrations agree that the 95th percentile of daily averaged COA con-

centrations at the different locations is 2–4 $\mu g\,m^{-3}$. For three other major cities, Manchester, Leeds and Birmingham, modelled annual average concentrations of COA were between 0.5–0.7 $\mu g\,m^{-3}$, but it should be noted that the model simulates the average concentration of the 5 km $\times$ 5 km grid cells, whereas it was shown for Manchester that cities can exhibit a central hot-spot of smaller scale (1–2 km in dimension). Therefore in some urban centres the contribution might be bigger than is modelled here.

The impact of COA concentrations is spatially very limited as the modelled concentrations drop markedly outside the highly populated urban areas. For example, the simulations estimated an annual average COA concentration of 0.12 $\mu g\,m^{-3}$ for the EMEP supersite Harwell (classified as rural background), which is ~80 km west of central London. This is comparable to estimates of COA concentrations at Harwell derived from a chemical mass balance (CMB) model applied to two weeks of measurements.

It is noted that it is possible that AMS-PMF measurements of COA concentrations might be overestimated by up to a factor of 2 (as was explained in Sect. 2.2). This means that the emission estimate of 7.4 Gg of COA per year (about 320 $mg\,person^{-1}\,day^{-1}$) could be a factor of 2 too high (but since COA is a primary PM emission, modelled COA concentrations scale linearly with changes in COA emission amount in the model). If this were the case then, depending on the degree of overestimation, COA would still an important contributor of PM in very central areas, but possibly less so in wider

urban or suburban areas.

In short, the spatially and temporally resolved COA emissions developed here for the UK can contribute to closing the gap between modelled and observed concentrations of carbonaceous aerosol and to total PM mass concentrations in urban areas.

*Acknowledgements.* The authors acknowledge the UK Department for Environment, Food and Rural Affairs (Defra) and the Devolved Administrations for funding aspects of the development of the EMEP4UK model (AQ0727), for partial support for the aerosol measurements,

as well as access to the AURN data, which were obtained from uk-air.defra.gov.uk and are subject to Crown 2014 copyright, Defra, licenced under the Open Government Licence (OGL). Partial support for the EMEP4UK modelling from the European Commission FP7 ECLAIRE project is gratefully acknowledged. This work was supported in part by the UK Natural Environment Research Council (NERC) ClearfLo project [grant ref. NE/H003169/1]. R. Ots was supported by a PhD studentship (University of Edinburgh and NERC-CEH contract 587/NEC03805). D. E. Young was supported by a NERC PhD studentship [ref. NE/I528142/1].





NCAR command language (NCL) was used to produce the maps (NCAR, 2015), and R, openair and ggplot2 for the analysis and all other plots (R Core Team, 2014; Carslaw and Ropkins, 2012; Wickham, 2009).



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





**Table 1.** Results of sensitivity tests for setting the annual total COA emission for the UK (gridded to workday population density). Model normalised mean biases of COA concentrations at the London Marylebone Road and North Kensington sites are shown for total UK emissions of 2 Gg, 8 Gg, and 7.4 Gg. A total emission of 7.4 Gg was chosen and is used in the rest of the simulations presented in this work.

| Site | Measured | Modelled (NMB) | | |
|---|---|---|---|---|
| | | 2 Gg | 8 Gg | 7.4 Gg |
| North Kensington | 0.8 µg m$^{-3}$ | -70% | +18% | +8% |
| Marylebone Road | 2.2 µg m$^{-3}$ | -75% | -2% | -4% |

**Table 2.** Measured and modelled mean concentrations of COA for approximately one year at two sites in London for weekdays (Monday–Friday) and weekends (Saturday–Sunday). Values in brackets are the 95% confidence interval of the mean. The number of (hourly) data points used for calculating each mean are given in Fig. 4.

| | Marylebone Road | | North Kensington | |
|---|---|---|---|---|
| | Meas. | Mod. | Meas. | Mod. |
| Weekdays [µg m$^{-3}$] | 2.2 (2.1–2.3) | 2.1 (2.0–2.2) | 0.7 (0.7–0.7) | 0.9 (0.9–1.0) |
| Weekend [µg m$^{-3}$] | 2.1 (2.0–2.3) | 2.2 (2.1–2.3) | 1.1 (1.0–1.2) | 0.9 (0.9–1.0) |

**Table 3.** Statistics for measured and modelled daily averaged COA concentrations at the two sites in London (site abbreviation as follows: MARY - Marylebone Road, NKEN - North Kensington). Up10 is the 90th percentile (upper 10% of the values), and Up5 is the 95th percentile (upper 5% of the values). The time-series of these values are shown in Fig. 5. Values in the "Modelled" line are for model values matched for data availability with the measurements. As Marylebone Road exhibits a few longer periods with missing measurements, modelled stats for the full year are also presented (red line in Fig. 5a). All units in µg m$^{-3}$.

| | | Mean | Median | Up10 | Up5 | Max. |
|---|---|---|---|---|---|---|
| MARY | Meas. | 2.2 | 2.1 | 3.5 | 4.1 | 5.9 |
| | Mod. | 2.1 | 1.8 | 3.2 | 3.9 | 10.0 |
| | Mod. (full year) | 2.0 | 1.8 | 3.1 | 3.7 | 10.0 |
| NKEN | Meas. | 0.8 | 0.6 | 1.7 | 2.0 | 4.1 |
| | Mod. | 0.9 | 0.7 | 1.4 | 2.0 | 6.8 |





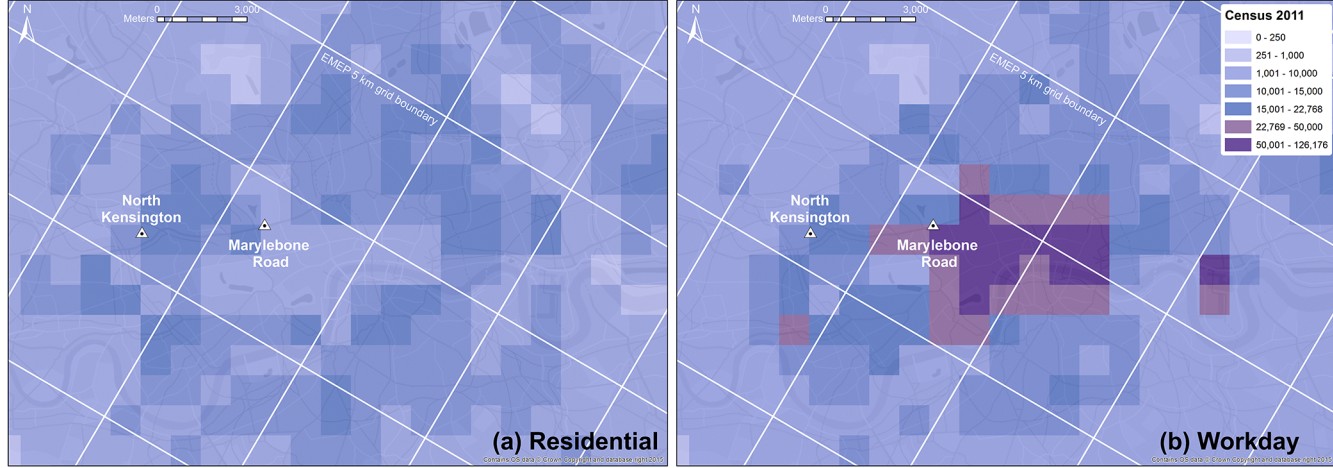

**Figure 1.** Residential (a) and workday (b) population density in central London at 1 km × 1 km resolution. The residential population maps are based on Reis et al. (2016). While the same methodology is applied to derive workday population maps, they are not yet published due to delays in the prevision of workday population census data for Scotland. Also shown are the measurement sites, and the EMEP4UK 5 km × 5 km grid used in this study (white lines). Underlying map contains Ordinance Survey (OS) data © Crown Copyright 2015.



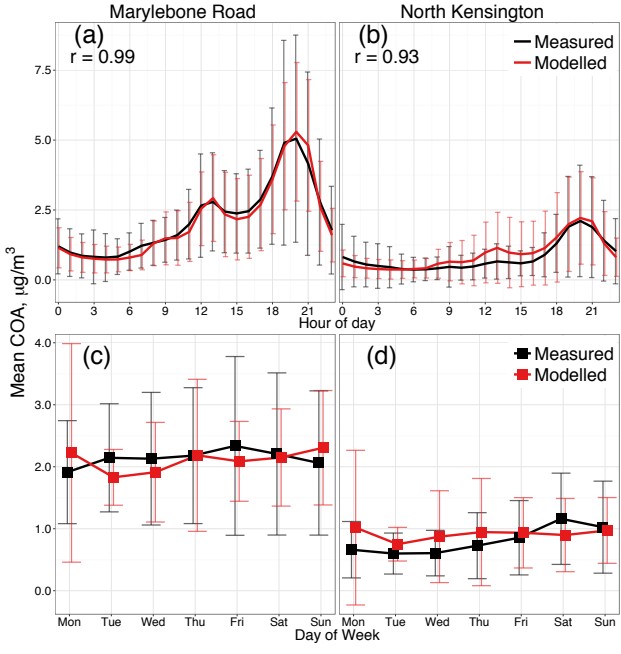

**Figure 2.** Average temporal profiles of COA concentrations at the two sites in central London in 2012: (a) diurnal profile at the Marylebone Road site, (b) diurnal profile at the North Kensington site, (c) day-of-week profile at the Marylebone Road site, (d) day-of-week profile at the North Kensington site. The timestamp of panels (a) and (b) is at the beginning of the hour. Also shown are standard deviations for each mean value.

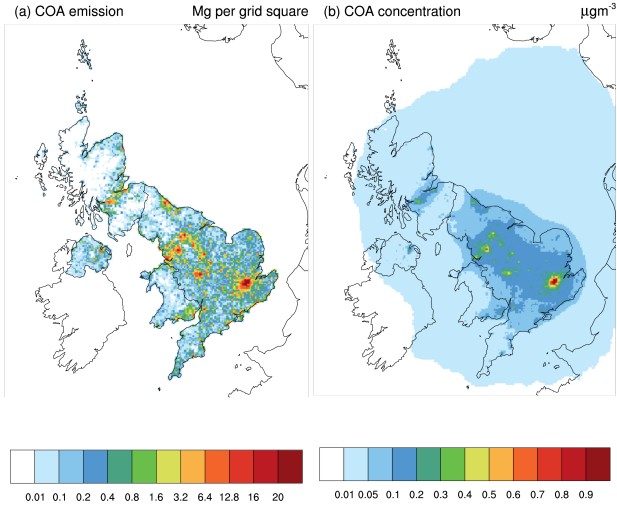

**Figure 3.** (a) Gridded COA emissions used in the model for the year 2012 (Mg per 5 km × 5 km grid cell, note the nonlinear scale), (b) annual average concentrations ($\mu g\,m^{-3}$).





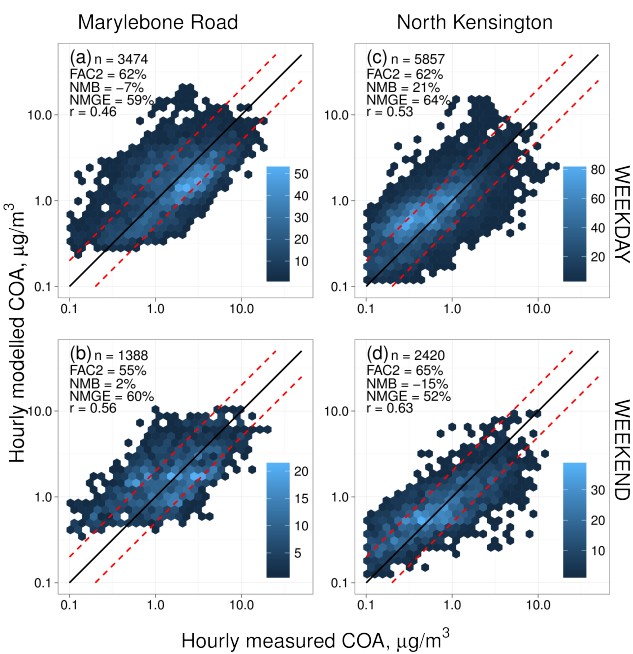

**Figure 4.** Data density scatterplots of measured versus modelled hourly COA concentrations for approximately one year at two sites in London: (a) Marylebone Road on weekdays, (b) Marylebone Road on weekends, (c) North Kensington on weekdays, (d) North Kensington on weekends. The colour scales indicate number of instances in a hexagonal (concentrations) bin. The straight lines are the 2:1, 1:1, and 1:2 lines. Note that on this Fig. the NMB for Marylebone Road for weekdays is -7%, but calculating the same statistic based on the numbers in Table 2 gives a NMB of -5%. This small discrepancy is caused by the rounding of concentrations for Table 2.





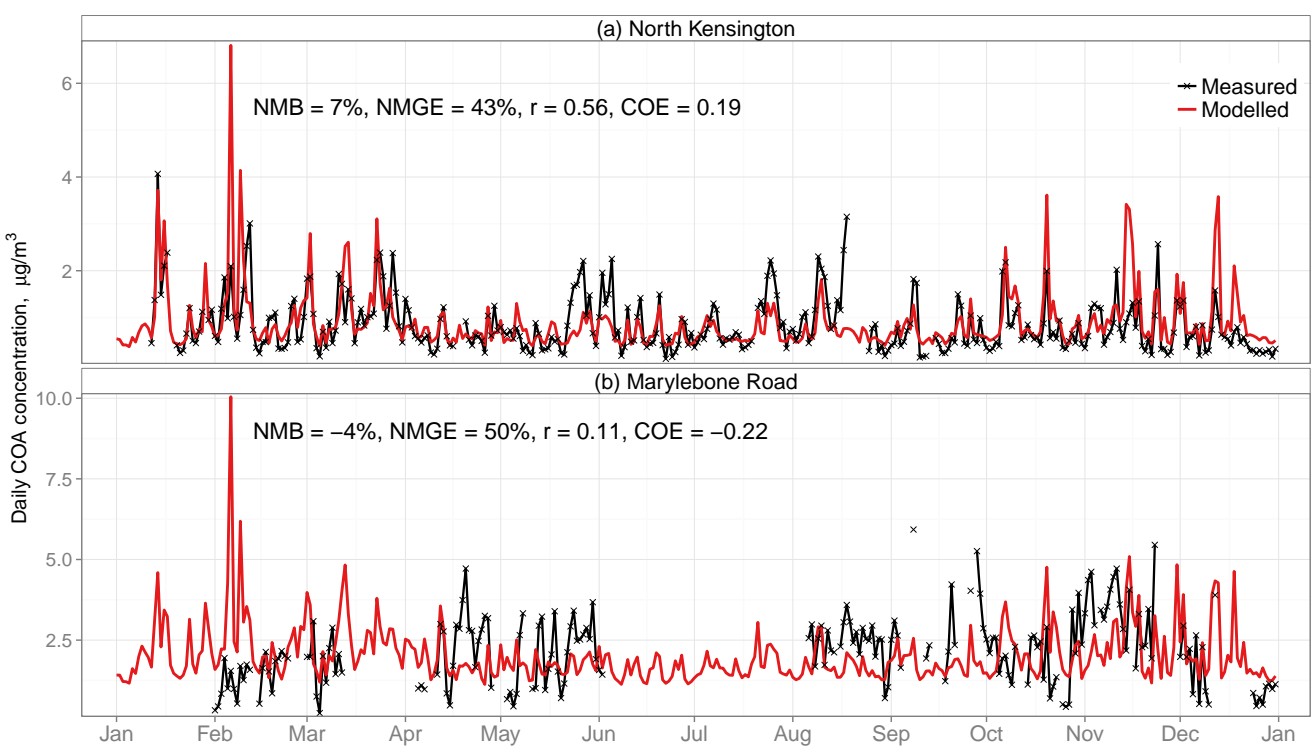

**Figure 5.** Time series of measured and modelled daily averaged COA concentrations at the (a) North Kensington, and (b) Marylebone Road measurement sites, year 2012.





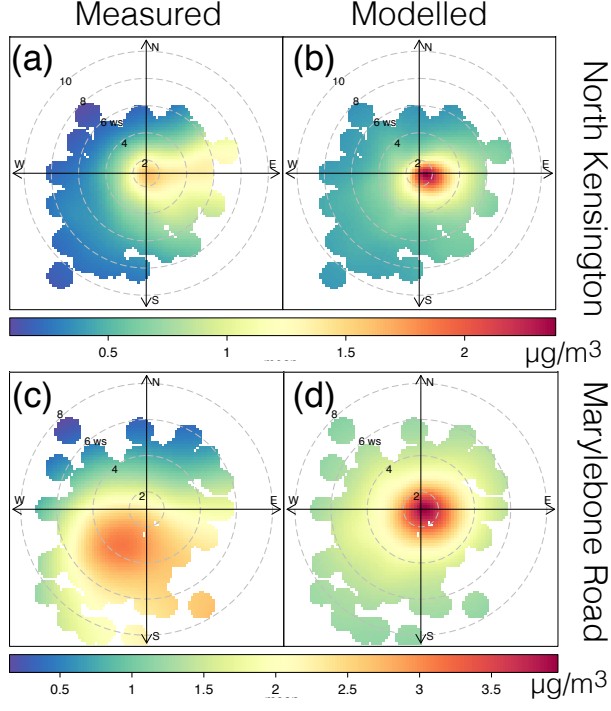

**Figure 6.** Polar plots of daily-average COA concentrations for wind speed (ws, $\mathrm{m\,s^{-1}}$) and direction measured at the Heathrow Airport meteorological station (Met Office, 2012). (a) measured and (b) modelled concentrations at the North Kensington site. (c) measured and (d) modelled concentrations at the Marylebone Road site.

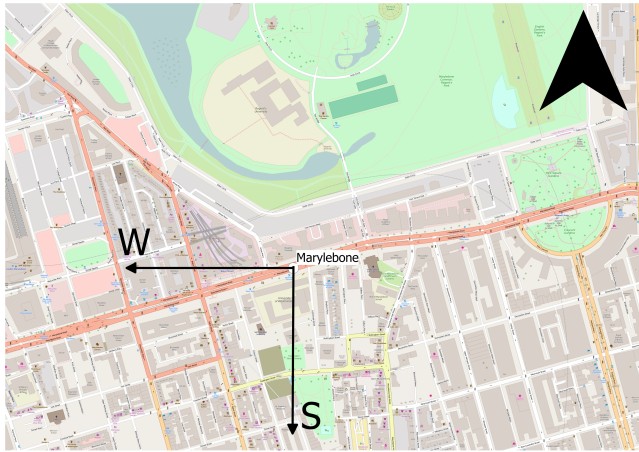

**Figure 7.** Location of the Marylebone Road measurement site, arrows indicate the West and South directions from the site. The measurement station is on the southern pavement of the street. Map from © OpenStreetMap contributors.





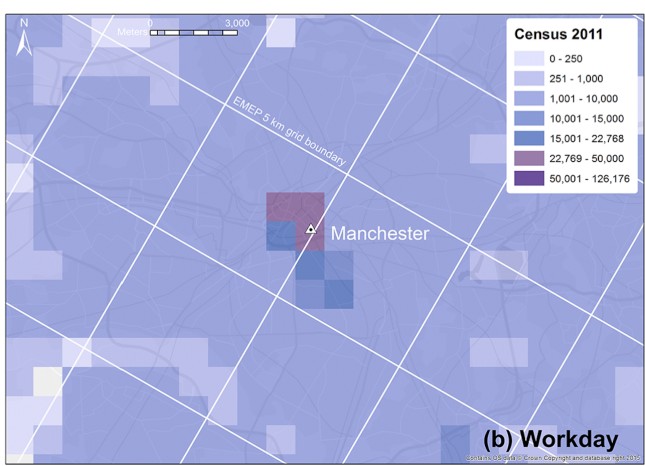

**Figure 8.** Workday population density in Manchester at 1 km × 1 km resolution in the OSGB36 (Ordinance Survey Great Britain 1936) projection. Also shown is the measurement site, and the EMEP4UK 5 km × 5 km grid used in this study (white lines). Underlying map from © OpenStreetMap contributors.





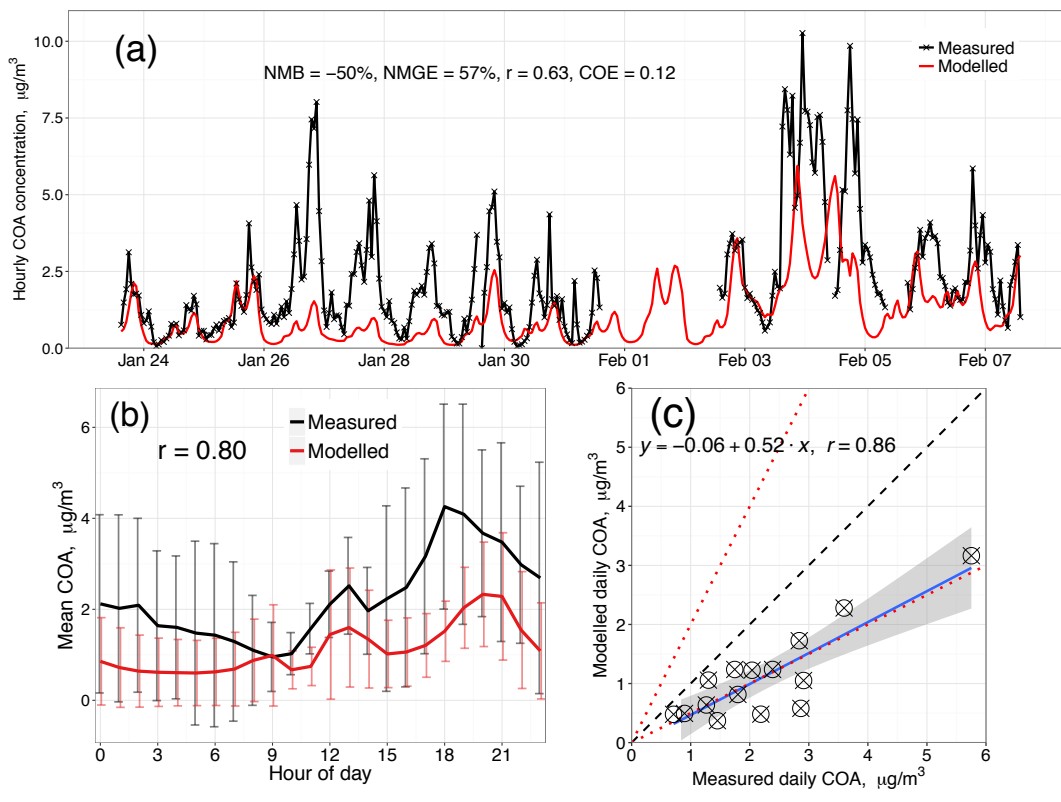

**Figure 9.** Comparison of modelled COA concentrations with an independent dataset of AMS measurements in Manchester, 2007. (a) Time series of measured and modelled hourly averaged COA concentrations. (b) Average diurnal profiles of measured and modelled COA (the timestamp is at the beginning of the hour, also shown are standard deviations for each mean value). (c) Scatterplots of daily-averaged modelled versus measured concentrations (the dotted and dashed lines are the 2:1, 1:1, and 1:2 lines, the blue line is the linear fit, the shading is the 95% confidence interval of the fit).





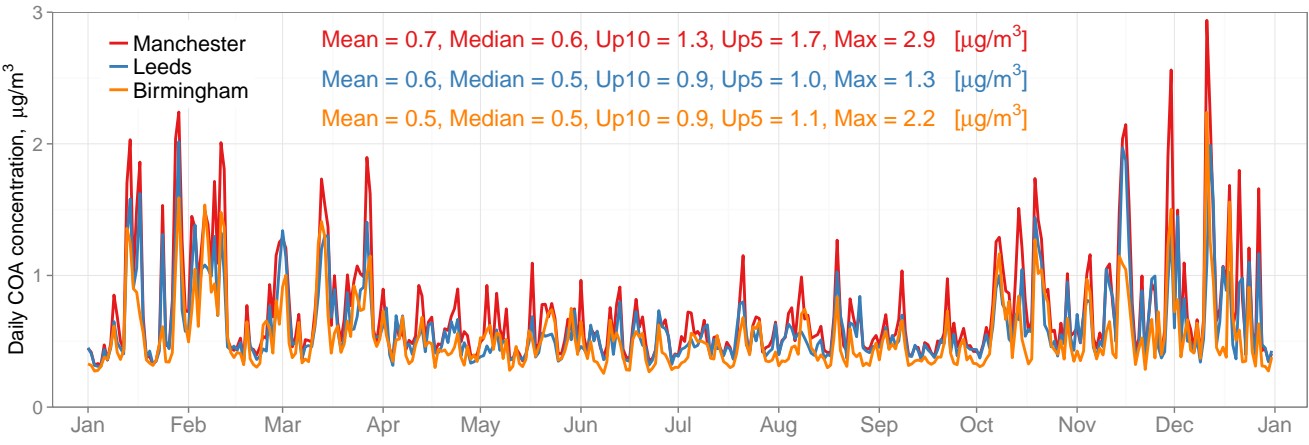

**Figure 10.** Time series of modelled daily-averaged COA concentrations for Manchester, Leeds, and Birmingham, year 2012. Up10 is the 90th percentile (upper 10% of the values), and Up5 is the 95th percentile (upper 5% of the values). The locations of these cities are shown in Fig. 11.

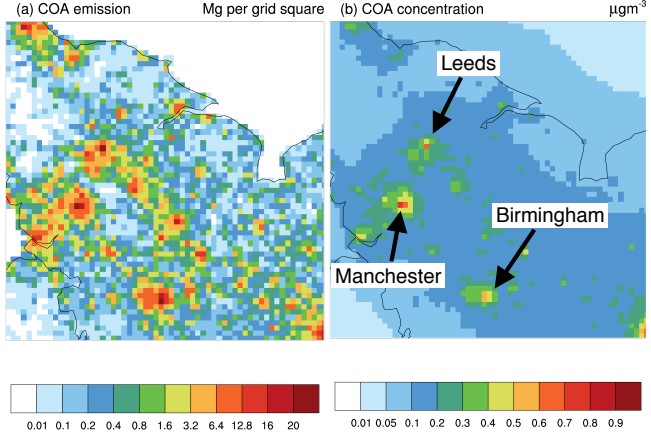

**Figure 11.** As Fig. 3, but zoomed in on northern England to show three other major cities with large estimated COA emissions: Manchester, Leeds, and Birmingham. (a) total COA emissions for the year 2012 (Mg per 5 km × 5 km grid cell, note the nonlinear scale), (b) annual average concentrations ($\mu\mathrm{g\,m^{-3}}$).



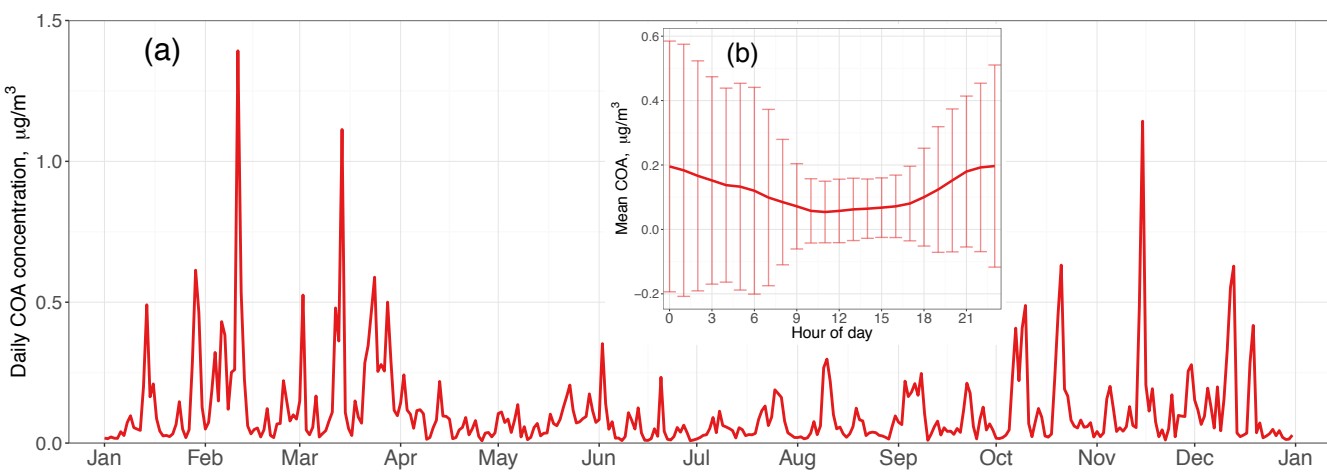

**Figure 12.** Modelled COA concentrations for the Harwell EMEP supersite location (a rural background site ~80 km from central London), year 2012. (a) Time series of modelled daily averaged COA concentrations. (b) Average diurnal profiles (the timestamp is at the beginning of the hour, also shown are standard deviations for each mean value).