# Peer review of "Model simulations of cooking organic aerosol (COA) over the UK using estimates of emissions based on measurements at two sites in London"

_Atmospheric Chemistry and Physics, 2016_

## Referee Comment (RC1) · Anonymous Referee #2 · 25 Jul 2016

This is a well written manuscript that describes an important contribution to the organic aerosol scientific literature. It describes a method that can be used to include food cooking primary organic aerosol emissions within regional chemical transport models. The method is well thought. The paper also clearly articulates the uncertainties in the measurement-based quantification of cooking organic aerosol. The paper describes the series of model runs that were developed to simulate the observations and thus derive a best-estimate of the cooking emissions. The tables and figures are of very good quality and easy to read and interpret.

I only have one technical correction and one suggested improvement.

Page 11, Line 4. Change "2,2" to "2.2"

The authors introduce the prior literature studies and use the COA/OA percent ratio as the metric to compare (Page 2, Line 22-32). It would be helpful to include this study's model and measurement estimate of the COA/OA percent ratio in the abstract, so that the research community can conveniently compare to other studies.

---

## Referee Comment (RC2) · Anonymous Referee #3 · 21 Aug 2016

This paper describes a development of a new top-down emission inventory for cooking organic aerosols (COA) in the UK. The COA emission estimates were included in an of-fline air quality model to simulate the COA distribution over a British Isles domain. The simulated COA concentrations were compared with surface measurements (inferred from PMF analysis) of COA concentrations at several cities in the UK.

It's crucial to quantify the contribution of the COA emissions to total OA burden. This also could help to interpret the radiocarbon analysis of particulate matter. Currently there is poor understanding of the COA emissions. As the authors noted this sector is totally missing from the European emission inventories of primary OA. The paper could help us to understand and quantify the role of COA emissions in air qualiy over major

metropolitan areas in Europe.

I think the text needs some revision for final publication.

Major comments: The paper neglects the discussion of radiocarbon analysis of organic carbon (OC). Was radiocarbon analysis done during the ClearfLo field campaign? I suggest adding discussion of such studies done in the past, e.g. Weber et al., 2007, Zotter et al., 2014. How better characterization of COA could help to explain the radiocarbon analysis of OC, such as modern vs. fossil carbon in aerosols?

There are a few studies, where the ambient COA was estimated using the PMF analysis. The paper by Hayes et al. 2015 reported COA, namely cooking influenced organic aerosol (CIOA) estimates based on the AMS measurements during the CalNex field campaign in the Los Angeles basin. Hayes et al., 2015 discuss the uncertainties related to identifying the OA burden due to the cooking sources. The findings of the study by Hayes et al., 2015 aren't discussed in this paper at all. Also, I think using the term as "CIOA" instead of "COA" would be more accurate.

The authors treated COA as non-volatile. Please elaborate on this point. What about other POA species, are they also treated as non-volatile. From the PMF analysis do you identify COA as HOA (primary) or OOA like (secondary)? Wouldn't scaling the COA emissions directly from the atmospheric measurements (by neglecting secondary OA) lead to overestimation of the COA emissions in this study?

Another missing point in the paper is the role of intermediate VOCs (IVOC) from the cooking sources. The studies by Schauer et al. aren't referenced here at all. I realize that it's hard to characterize the emissions of the IVOCs from cooking sources. But discussing on this topic is important here.

The authors mention possible measurement uncertainties in the text, but I don't see any uncertainty numbers related to the collection efficiency and PMF method are presented in this paper. How much those measurement uncertainties change the conclusions

drawn from the model-measurement comparisons?

Paragraph 20: I think this statement is little misleading. At present there are a number of SOA precursors and mechanisms (proposed during last 5-10 years) that are used in the models, which lead to overestimation of OA in some cases. I think what is more important now to constrain the different mechanisms (aging e.g.) and sources of OA in air quality models.

It'd help to include the measured total OA concentrations at the sites discussed in this paper. Also, their PMF composition to give a better idea to a reader about the role of COA and other sources in driving OA pollution across those cities.

References: Weber, R. J., A. P. Sullivan, R. E. Peltier, A. Russell, Y. Bo, Z. Mei, J. de Gouw, C. Warneke, C. Brock, J. S. Holloway, E. L. Atlas and E. Edgerton (2007). "A study of secondary organic aerosol formation in the anthropogenic-influenced southeastern United States." Journal of Geophysical Research-Part D-Atmospheres 112(13): D13213-13211-D13213-D13213-13213.

Zotter, P., El-Haddad, I., Zhang, Y., Hayes, P. L., Zhang, X., Lin, Y.-H., Wacker, L., Schnelle-Kreis, J., Abbaszade, G., Zimmermann, R., Surratt, J. D., Weber, R., Jimenez, J. L., Szidat, S.,Baltensperger, U., and Prévôt, A. S. H.: Diurnal cycle of fossil and nonfossil carbon using radiocarbon analyses during CalNex, J. Geophys. Res.-Atmos., 119, 6818–6835, 2014.

Hayes, P. L., A. G. Carlton, K. R. Baker, R. Ahmadov, R. A. Washenfelder, S. Alvarez, B. Rappengluck, J. B. Gilman, W. C. Kuster, J. A. de Gouw, P. Zotter, A. S. H. Prevot, S. Szidat, T. E. Kleindienst, J. H. Offenberg, P. K. Ma and J. L. Jimenez (2015). "Modeling the formation and aging of secondary organic aerosols in Los Angeles during CalNex 2010." Atmospheric Chemistry and Physics 15(10): 5773-5801.

Schauer, J. J., M. J. Kleeman, G. R. Cass and B. R. T. Simoneit (1999). "Measurement of emissions from air pollution sources. 1. C-1 through C-29 organic compounds from

meat charbroiling." Environmental Science & Technology 33(10): 1566-1577.

---

## Author Comment (AC1) · 27 Sep 2016

**acp-2016-342: Model simulations of cooking organic aerosol (COA) over the UK using estimates of emissions based on measurements at two sites in London**

We thank the reviewer for their very supportive comments. We respond to each comment individually below. The reviewer's comments are in italics and blue font, our responses are in normal text.

**Anonymous Referee #2**

*This is a well written manuscript that describes an important contribution to the organic aerosol scientific literature. It describes a method that can be used to include food cooking primary organic aerosol emissions within regional chemical transport models. The method is well thought. The paper also clearly articulates the uncertainties in the measurement-based quantification of cooking organic aerosol. The paper describes the series of model runs that were developed to simulate the observations and thus derive a best-estimate of the cooking emissions. The tables and figures are of very good quality and easy to read and interpret.*

*I only have one technical correction and one suggested improvement.*
*Page 11, Line 4. Change "2,2" to "2.2"*

Response: Done.

*The authors introduce the prior literature studies and use the COA/OA percent ratio as the metric to compare (Page 2, Line 22-32). It would be helpful to include this study's model and measurement estimate of the COA/OA percent ratio in the abstract, so that the research community can conveniently compare to other studies.*

Response: Agreed, we have added the percentage contribution of our COA to total measured OA to the abstract text:
"The modelled annual average contribution of COA to ambient particulate matter (PM) in central London was between 1–2 µg m$^{-3}$ **(~20% of total measured OA1)** …"

---

## Author Comment (AC2) · 27 Sep 2016

**acp-2016-342: Model simulations of cooking organic aerosol (COA) over the UK using estimates of emissions based on measurements at two sites in London**

We thank the reviewer for their supportive comments. We respond to each comment individually below. The reviewer's comments are in italics and blue font, our responses are in normal text.

**Anonymous Referee #3**

*This paper describes a development of a new top-down emission inventory for cooking organic aerosols (COA) in the UK. The COA emission estimates were included in an offline air quality model to simulate the COA distribution over a British Isles domain. The simulated COA concentrations were compared with surface measurements (inferred from PMF analysis) of COA concentrations at several cities in the UK. It's crucial to quantify the contribution of the COA emissions to total OA burden. This also could help to interpret the radiocarbon analysis of particulate matter. Currently there is poor understanding of the COA emissions. As the authors noted this sector is totally missing from the European emission inventories of primary OA. The paper could help us to understand and quantify the role of COA emissions in air quality over major metropolitan areas in Europe.*

*I think the text needs some revision for final publication.*

*Major comments: The paper neglects the discussion of radiocarbon analysis of organic carbon (OC). Was radiocarbon analysis done during the ClearfLo field campaign? I suggest adding discussion of such studies done in the past, e.g. Weber et al., 2007, Zotter et al., 2014. How better characterization of COA could help to explain the radiocarbon analysis of OC, such as modern vs. fossil carbon in aerosols?*

Response: Radiocarbon (i.e. 14C) is certainly an excellent tracer for distinguishing between fossil and contemporary carbon. However, without additional measurement data, and usually some assumptions about source emission ratios, radiocarbon cannot directly distinguish the sources of the fossil or contemporary carbon or whether the carbon is of primary or secondary origin (Heal, 2014). The radiocarbon in some samples of PM2.5 was determined during the ClearfLo campaign yielding average proportions for non-fossil TC of 53% at the North Kensington urban background site and 64% at a rural background site at Detling to the east (and normally downwind) of London (Crilley et al., 2015). Whilst these radiocarbon data are only for a very small subset of time within the full year of ClearfLo campaign covered in our modelling work, the greater than one-half contribution of non-fossil carbon to TC in London is in line with similar proportions of non-fossil carbon in PM reported in Birmingham, UK (Heal et al., 2011) and in urban airsheds elsewhere (for example, the Weber et al. (2007) and Zotter et al. (2014) studies the reviewer highlights above). Where reported, non-fossil contributions to OC are higher than for TC (Heal, 2014); for example mean non-fossil contribution to OC in Birmingham, UK, was 76% (Heal et al., 2011). The carbon in COA will be non-fossil (being derived from the food itself and from biologically-derived cooking oils). Therefore, the finding in our current study that COA can be a notable component in urban air (we report ~20% of $OA_1$ in central London) is entirely in line with radiocarbon data, including the estimate by

Zotter et al. (2014) that cooking contributed at least 25% to non-fossil OC in Los Angeles air. (If OA in London was ~76% non-fossil, as in Birmingham, then an estimated contribution of ~20% cooking to OA in London would be equivalent to ~26% cooking contribution to non-fossil OA.) Without detailed emissions data for all the other fossil and non-fossil carbon sources that can contribute to OA (and in particular there are known shortcomings in the European emissions inventories for domestic wood burning), it is not possible to use the model for detailed quantitative apportionment of fossil/non-fossil OC beyond the observations made above that the findings from the model work are entirely consistent with conclusions derived from radiocarbon measurements.

Given that it is not possible to make detailed quantitative comparisons between model output (for a year of study) and short-duration radiocarbon measurements we cannot elaborate much on this in our paper. However, we acknowledge the validity of including some discussion of these matters so we have now added the following text as a new paragraph at the end of Section 3.2:

"Table 2 and Figure 5 show that the modelled annual average contribution of COA to ambient particulate matter (PM) in central London was between 1-2 $\mu$g m$^{-3}$, which corresponds to ~20% of OA$_1$. The carbon in COA will be non-fossil, being derived from the food itself and from biologically-derived cooking oils. Using radiocarbon (carbon-14) measurements on some daily samples of PM$_{2.5}$ collected during the ClearfLo campaign, the average non-fossil contributions to total carbon (TC) at the North Kensington urban background site, and at the Detling rural background site east of London, were determined to be 53% and 64% on average, respectively (Crilley et al., 2015). The greater than one-half contribution of non-fossil carbon to TC in London is in line with similar proportions of non-fossil carbon in PM reported in Birmingham, UK (Heal et al., 2011) and in urban airsheds elsewhere (e.g. Weber et al., 2007; Zotter et al., 2014). Where reported, non-fossil contributions to OC are higher than for TC (Heal, 2014); for example mean non-fossil contribution to OC in Birmingham, UK, was 76% (Heal et al., 2011). Therefore the finding here that COA can be a notable component in urban air is entirely consistent with radiocarbon apportionments, including the estimate by Zotter et al. (2014) that cooking contributed at least 25% to non-fossil OC in Los Angeles air."

*There are a few studies, where the ambient COA was estimated using the PMF analysis. The paper by Hayes et al. 2015 reported COA, namely cooking influenced organic aerosol (CIOA) estimates based on the AMS measurements during the CalNex field campaign in the Los Angeles basin. Hayes et al., 2015 discuss the uncertainties related to identifying the OA burden due to the cooking sources. The findings of the study by Hayes et al., 2015 aren't discussed in this paper at all. Also, I think using the term as "CIOA" instead of "COA" would be more accurate.*

Response: We thank the reviewer for bringing this work to our attention. We have added the following text to Sect. 2.2 (AMS measurements used in this study). Note that we actually cite to Hayes et al. 2013, as this is where the original observations are described, whereas Hayes et al. 2015 just cited to the earlier paper.

"Furthermore, Hayes et al. (2013) observed that the correlation between HOA+COA and CO is stronger than the correlation between just HOA and CO (0.71 vs 0.59). They speculated this

could mean the COA component identified may also include some particulate mass from non-cooking sources such as traffic."

While accepting the reviewer may have a valid point that the term CIOA rather than COA might be a more accurate descriptor for this PMF factor, we have chosen to continue using COA here because the measurement datasets we have used in this study (Young et al. 2015; Allan et al. 2010) have identified this source with this descriptor.

*The authors treated COA as non-volatile. Please elaborate on this point. What about other POA species, are they also treated as non-volatile. From the PMF analysis do you identify COA as HOA (primary) or OOA like (secondary)? Wouldn't scaling the COA emissions directly from the atmospheric measurements (by neglecting secondary OA) lead to overestimation of the COA emissions in this study?*

Response: The PMF analysis identifies COA as a primary component. The COA that has gone through atmospheric ageing after emissions would be included under the PMF OOA factors, but as we only used the (primary) COA factor our emission estimate might be an underestimate. However, as we based our estimates on sites in central London (i.e. a high source region), this underestimation of COA that has already 'transformed' to OOA is likely to be small.

The volatility of POA species is a major source of uncertainty in atmospheric chemistry transport models. In acknowledgement of this we have now added the following to Sect. 2.1 (Model description):

"Treating POA as non-volatile is a simplification as, in reality, some POA may evaporate on atmospheric dilution, go through atmospheric ageing, and condense back into the particulate phase thus becoming secondary OA (SOA; Robinson et al. (2007)). The volatility distribution and ageing rates are, however, still a major source of uncertainty in atmospheric chemistry models (Ots et al. (2016) and references therein)."

*Another missing point in the paper is the role of intermediate VOCs (IVOC) from the cooking sources. The studies by Schauer et al. aren't referenced here at all. I realize that it's hard to characterize the emissions of the IVOCs from cooking sources. But discussing on this topic is important here.*

Response: We have now added the following text to the model description section immediately after the additional insertion of text on POA referred to in the previous comment:

"Furthermore, some POA emissions are accompanied by emissions of intermediate volatility organic compounds (IVOCs; e.g. Shrivastava et al. (2008) based on Schauer et al. (1999)), but to our knowledge, there are currently no measurements or estimates of cooking-IVOCs to use as a basis for modelling."

*The authors mention possible measurement uncertainties in the text, but I don't see any uncertainty numbers related to the collection efficiency and PMF method are presented in this paper. How much those measurement uncertainties change the conclusions drawn from the model-measurement comparisons?*

Response: We do state, in a couple of places, that the uncertainty (most likely overestimation) in quantification of the contribution of COA derived from PMF of AMS data is up to a factor of 2. First, in Sect. 2.2 (AMS studies used in this study):

"Nevertheless, there are some inherit uncertainties involved in deriving COA concentrations with AMS measurements. For example, AMS measurements need to be corrected for the fraction of aerosol that is not effectively vaporised due to bounce from the hot surface involved in the AMS's detection mechanism. Whilst this is well characterised for typical, internally-mixed ambient aerosols (e.g. Middlebrook et al. (2012)), it is possible that the COA measured by the AMS is not well mixed with other aerosol components and could therefore be detected at a higher efficiency. If this were the case, AMS measurements may overestimate COA concentrations by up to a factor of 2."

Secondly, at the end of the Conclusions:

"It is noted that it is possible that AMS-PMF measurements of COA concentrations might be overestimated by up to a factor of 2 (as was explained in Sect. 2.2). This means that the emission estimate of 7.4 Gg of COA per year (about 320 mg person$^{-1}$ day$^{-1}$) could be a factor of 2 too high (but since COA is a primary PM emission, modelled COA concentrations scale linearly with changes in COA emission amount in the model). If this were the case then, depending on the degree of overestimation, COA would still an important contributor of PM in very central areas, but possibly less so in wider urban or suburban areas."

*Paragraph 20: I think this statement is little misleading. At present there are a number of SOA precursors and mechanisms (proposed during last 5-10 years) that are used in the models, which lead to overestimation of OA in some cases. I think what is more important now to constrain the different mechanisms (aging e.g.) and sources of OA in air quality models.*

Response: Whilst we agree that significant improvements in SOA precursor emissions and modelling have been achieved, our work was focused on a specific primary component (COA) that has been identified by several AMS studies as such. Some studies have achieved an overestimation of total OA by increasing SOA precursors, but with the dataset we are using (so not just total OA, but PMF apportionment of different components of OA) it is clear that primary OA from cooking sources is of importance to total OA (and therefore to total PM).

*It'd help to include the measured total OA concentrations at the sites discussed in this paper. Also, their PMF composition to give a better idea to a reader about the role of COA and other sources in driving OA pollution across those cities.*

Response: Agreed, we have added the following sentences to Sect. 2.2 (AMS measurements used in this study):

"Annual average $OA_1$ during 2012 at the Marylebone Road site was measured at 8.5 µg m$^{-3}$: 0.8 µg m$^{-3}$ SFOA (9%), 3.0 µg m$^{-3}$ SOA (36%), 2.5 µg m$^{-3}$ HOA (29%), and 2.2 µg m$^{-3}$ COA (26%)." and
"Annual average $OA_1$ during 2012 at the North Kensington site was measured at 4.2 µg m$^{-3}$: 1.0 µg m$^{-3}$ SFOA (24%), 1.6 µg m$^{-3}$ SOA (38%), 0.8 µg m$^{-3}$ HOA (19%), and 0.8 µg m$^{-3}$ COA (19%)."

*References cited by reviewer*
*Weber, R. J., A. P. Sullivan, R. E. Peltier, A. Russell, Y. Bo, Z. Mei, J. de Gouw, C. Warneke, C. Brock, J. S. Holloway, E. L. Atlas and E. Edgerton (2007). "A study of secondary organic aerosol formation in the anthropogenic-influenced southeastern United States." Journal of Geophysical Research-Part D-Atmospheres 112(13): D13302.*

*Zotter, P., El-Haddad, I., Zhang, Y., Hayes, P. L., Zhang, X., Lin, Y.-H., Wacker, L., Schnelle Kreis, J., Abbaszade, G., Zimmermann, R., Surratt, J. D.,Weber, R., Jimenez, J. L., Szidat, S.,Baltensperger, U., and Prévôt, A. S. H.: Diurnal cycle of fossil and nonfossil carbon using radiocarbon analyses during CalNex, J. Geophys. Res.-Atmos., 119, 6818–6835, 2014.*

*Hayes, P. L., A. G. Carlton, K. R. Baker, R. Ahmadov, R. A. Washenfelder, S. Alvarez, B. Rappengluck, J. B. Gilman, W. C. Kuster, J. A. de Gouw, P. Zotter, A. S. H. Prevot, S. Szidat, T. E. Kleindienst, J. H. Offenberg, P. K. Ma and J. L. Jimenez (2015). "Modeling the formation and aging of secondary organic aerosols in Los Angeles during CalNex 2010." Atmospheric Chemistry and Physics 15(10): 5773-5801.*

*Schauer, J. J., M. J. Kleeman, G. R. Cass and B. R. T. Simoneit (1999). "Measurement of emissions from air pollution sources. 1. C-1 through C-29 organic compounds from meat charbroiling." Environmental Science & Technology 33(10): 1566-1577.*

References cited in our responses:
Allan, J. D., Williams, P. I., Morgan, W. T., Martin, C. L., Flynn, M. J., Lee, J., Nemitz, E., Phillips, G. J., Gallagher, M. W., and Coe, H.: Contributions from transport, solid fuel burning and cooking to primary organic aerosols in two UK cities, Atmos. Chem. Phys., 10, 647–668, doi:10.5194/acp-10-647-2010, 2010.

Crilley, L. R., Bloss, W. J., Yin, J., Beddows, D. C. S., Harrison, R. M., Allan, J. D., Young, D. E., Flynn, M., Williams, P., Zotter, P., Prevot, A. S. H., Heal, M. R., Barlow, J. F., Halios, C. H., Lee, J. D., Szidat, S. and Mohr, C.: Sources and contributions of wood smoke during winter in London: assessing local and regional influences, Atmos. Chem. Phys., 15, 3149-3171, 2015.

Hayes, P. L., Ortega, A. M., Cubison, M. J., Froyd, K. D., Zhao, Y., Cliff, S. S., Hu, W. W., Toohey, D. W., Flynn, J. H., Lefer, B. L., Grossberg, N., Alvarez, S., Rappenglück, B., Taylor, J. W., Allan, J. D., Holloway, J. S., Gilman, J. B., Kuster, W. C., de Gouw, J. A., Massoli, P., Zhang, X., Liu, J., Weber, R. J., Corrigan, A. L., Russell, L. M., Isaacman, G., Worton, D. R., Kreisberg, N. M., Goldstein, A. H., Thalman, R., Waxman, E. M., Volkamer, R., Lin, Y. H., Surratt, J. D., Kleindienst, T. E., Offenberg, J. H., Dusanter, S., Griffith, S., Stevens, P. S., Brioude, J., Angevine, W. M., and Jimenez, J. L.: Organic aerosol composition and sources in Pasadena, California, during the 2010 CalNex campaign, Journal of Geophysical Research:

Atmospheres, 118, 9233–9257, doi:10.1002/jgrd.50530, http://onlinelibrary.wiley.com/doi/10.1002/jgrd.50530/abstract, 2013.

Heal, M. R.: The application of carbon-14 analyses to characterisation of sources of atmospheric carbonaceous particulate matter: a review, Anal. Bioanal. Chem., 406, 81-98, 2014.

Heal, M. R., Naysmith, P., Cook, G. T., Xu, S., Raventos Duran, T. and Harrison, R. M.: Application of 14C analyses to source apportionment of carbonaceous PM2.5 in the UK, Atmos. Environ., 45, 2341-2348, 2011.

Ots, R., Young, D. E., Vieno, M., Xu, L., Dunmore, R. E., Allan, J. D., Coe, H., Williams, L. R., Herndon, S. C., Ng, N. L., Hamilton, J. F., Bergström, R., Di Marco, C., Nemitz, E., Mackenzie, I. A., Kuenen, J. J. P., Green, D. C., Reis, S., and Heal, M. R.: Simulating secondary organic aerosol from missing diesel-related intermediate-volatility organic compound emissions during the Clean Air for London (ClearfLo) campaign, Atmos. Chem. Phys., 16, 6453-6473, doi:10.5194/acp-16-6453-2016, 2016.

Schauer, J. J., Kleeman, M. J., Cass, G. R., and Simoneit, B. R. T.: Measurement of Emissions from Air Pollution Sources. 2. C1 through C30 Organic Compounds from Medium Duty Diesel Trucks, Environmental Science & Technology, 33, 1578–1587, doi:10.1021/es980081n, http://dx.doi.org/10.1021/es980081n, 1999.

Shrivastava, M. K., Lane, T. E., Donahue, N. M., Pandis, S. N., and Robinson, A. L.: Effects of gas particle partitioning and aging of primary emissions on urban and regional organic aerosol concentrations, Journal of Geophysical Research-Atmospheres, 113, doi:10.1029/2007jd009735, 2008.

Young, D. E., Allan, J. D., Williams, P. I., Green, D. C., Flynn, M. J., Harrison, R. M., Yin, J., Gallagher, M. W., and Coe, H.: Investigating the annual behaviour of submicron secondary inorganic and organic aerosols in London, Atmos. Chem. Phys., 15, 6351–6366, doi:10.5194/acp-15-6351-2015, 2015.

---

## Author Response (AR2)

Dear Editor,

Thank you very much for your prompt decision, as well as for bringing the two very recent and relevant references to our attention. We have included both, the sentences are highlighted below.

[revised manuscript text omitted]